# Transferring chemical and energetic knowledge between molecular systems with machine learning

Sajjad Heydari [1,5], Stefano Raniolo[2,5], Lorenzo Livi[1,3 ✉] & Vittorio Limongelli[2,4 ✉]

Predicting structural and energetic properties of a molecular system is one of the fundamental tasks in molecular simulations, and it has applications in chemistry, biology, and medicine. In the past decade, the advent of machine learning algorithms had an impact on molecular simulations for various tasks, including property prediction of atomistic systems. In this paper, we propose a novel methodology for transferring knowledge obtained from simple molecular systems to a more complex one, endowed with a significantly larger number of atoms and degrees of freedom. In particular, we focus on the classification of high and low free-energy conformations. Our approach relies on utilizing (i) a novel hypergraph representation of molecules, encoding all relevant information for characterizing multi-atom interactions for a given conformation, and (ii) novel message passing and pooling layers for processing and making free-energy predictions on such hypergraph-structured data. Despite the complexity of the problem, our results show a remarkable Area Under the Curve of 0.92 for transfer learning from tri-alanine to the deca-alanine system. Moreover, we show that the same transfer learning approach can also be used in an unsupervised way to group chemically related secondary structures of deca-alanine in clusters having similar free-energy values. Our study represents a proof of concept that reliable transfer learning models for molecular systems can be designed, paving the way to unexplored routes in prediction of structural and energetic properties of biologically relevant systems.

[1] Department of Computer Science, University of Manitoba, Winnipeg, MB R3T 2N2, Canada. [2] Faculty of Biomedical Sciences, Euler Institute, Università della Svizzera italiana (USI), via G. Buffi 13, CH-6900 Lugano, Switzerland. [3] Department of Computer Science, University of Exeter, Exeter EX4 4QF, UK. [4] Department of Pharmacy, University of Naples "Federico II", via D. Montesano 49, I-80131 Naples, Italy. [5]These authors contributed equally: Sajjad Heydari and Stefano Raniolo. ✉email: lorenzo.livi@umanitoba.ca; vittoriolimongelli@gmail.com

Molecular simulations are nowadays a fundamental field of investigation in applied sciences, from chemistry to biology and medicine[1–4]. They are typically used to predict the properties of a system with relatively good accuracy. In the era of artificial intelligence and machine learning (ML), new challenges are posed in this field, trying to exploit the ability of ML algorithms to deal with a large amount of data and extrapolate importantly, yet not immediately apparent information for the system under investigation. ML techniques have been indeed applied to chemo-informatics problems—prediction of compounds properties like solubility, toxicity, etc.—thanks to the relative abundance of experimental data[5,6]. In the last decade, the first attempts to employ ML in molecular simulations have also appeared. In particular, ML has been used to predict atomistic properties in molecular systems[7–11], also using first principle calculations (i.e., quantum mechanics)[12–14], and, more recently, in the identification of free-energy states and slow degrees of freedom in molecular systems[15,16]. However, the wealth of data represents a major limitation for ML applications and despite the increasing computing power, the sampling capability of a system's phase space still represents a hindering factor in all ML applications to molecular simulations. The sampling issue is even more evident in biologically relevant macromolecules made by thousands of atoms, like DNA and protein systems. In fact, despite a solid theory based on statistical mechanics[17], the large size of real molecules and the long timescale of the events under consideration, impede even the most advanced simulation techniques to study macromolecules in realistic conditions. A clear example is drug discovery, where the drug in vivo efficacy is determined by ligand-target binding kinetics (quantified as drug residence time[18–21]), which is hardly predictable by current simulation methods[22]. In fact, the free-energy landscapes of drug–protein interaction are typically characterized by a number of high barriers that separate various metastable states, trapping the simulation in limited parts of the energy landscape for extended periods of time[23]. Developing enhanced sampling techniques and coarse-grained representations[22,24–28] has significantly ameliorated the sampling capability. However, that remains insufficient in most of the real cases, characterized by the complex, long-timescale evolution of the system. As a result, the identification of the most probable, fundamental free-energy states is not feasible.

In order to overcome such a limitation, an attracting strategy consists of transferring the knowledge acquired on simple, computationally affordable systems to a much more complex one for predicting relevant properties of the complex system. This strategy is known by the name of transfer learning[29,30], and represents a rather unexplored field of investigations in molecular simulations so far[31]. Here, we address this challenge and propose a novel methodology based on transfer learning that allows learning the free-energy of a given molecular system—i.e., accurate free-energy data obtained from atomistic simulations—and transfer such information on a previously unseen molecular system of different size having a significantly larger number of atoms and degrees of freedom that cannot be easily characterized by the free-energy calculations. In particular, we aimed at the classification of low and high free-energy conformations. As shown in Fig. 1, the proposed methodology is based on a novel hypergraph representation of molecules introduced here, which allows encoding all the relevant information for characterizing the multi-atom interactions in a given conformation. The free-energy is then predicted by a novel neural network model capable of processing such hypergraphs as inputs. Although the literature already contains a few methods based on neural networks for processing hypergraphs[32–35] and simplicial complexes[36], such methods have some restrictions, e.g., they assume scalars as features for

hyperedges and do not offer pooling mechanisms for variable-size inputs, and therefore they are not suitable for the hypergraph representation of molecules introduced here.

We demonstrate the ability of the proposed hypergraph neural network (HNN) on a set of transfer learning experiments. The first one is performed on alanine dipeptide, with the aim to make predictions on the free-energy of a slightly more complex system given by the composition of three alanine peptides, called tri-alanine. Then, we move to a more challenging setting where transfer learning is performed between relatively simple systems (i.e., alanine dipeptide and tri-alanine) and a composition of ten alanine structures, called deca-alanine. This experimental setting represents a real case study since deca-alanine assumes secondary structures which are not present neither in alanine nor in tri-alanine. That is, the most probable conformations of the system, expressed as $\phi$ and $\psi$ torsion angles of alanine, are different in deca-alanine with respect to those assumed in alanine dipeptide and tri-alanine. Here, we show a remarkable classification performance, quantified by an Area Under the Curve (AUC) of 0.92. We also show that the same transfer learning approach can be used in an unsupervised way to group chemically related secondary structures of deca-alanine in clusters having similar free-energy values.

Our work is a proof of concept that it is possible, by means of a purposely built machine learning model, to predict free-energy values of a complex molecule using free-energy and structural data of a smaller, yet chemically related molecule, thus *de facto* overcoming the sampling issue for large systems.

## Results

**Molecular representation and processing**. The very first challenge in employing ML to study molecular systems is to develop a reliable molecular representation that is amenable to processing via ML algorithms. Two important properties that are desirable for molecule representations are uniqueness and invertibility[37]. Uniqueness means that each molecular structure is associated with a single representation; invertibility means that each representation is associated with a single molecule, hence giving rise to a one-to-one mapping. Most representations used for molecular generation are invertible, but some are not unique[38–41]. There are several reasons for non-uniqueness, including the representation not being invariant to the underlying physical symmetries of rotation, translation, and permutation of atomic indexes. While machine learning algorithms may be directly applied to physical 3D coordinates of atoms, it is preferable removing invariances by creating a more compact representation (removing degrees of freedom) and thus developing a unique representation for each molecule based on internal coordinates only.

Moreover, to be effective for the task at hand, the representation needs to encode both the structural and the physico-chemical properties of the system under investigation. Typically, multi-atom interactions are assessed by computing the potential energy $E_p$ of a structure[42], which is classically modeled[43] as the sum of four parts:

$$E_p = E_{bond} + E_{non-bond} + E_{angle} + E_{dihedral} \qquad (1)$$

This implies that $E_p(1)$ cannot be described by only accounting for the interaction between pairs of atoms (dependency on bond length, $E_{bond}$, and electrostatic interactions, $E_{non-bond}$). In fact, the potential energy contains terms that account for angles, $E_{angle}$, and dihedrals $E_{dihedral}$ (i.e., the angle formed by two planes defined by four atoms), which are determined by considering the interaction of three and four atoms, respectively. Accordingly, the commonly used graph representations for molecules[39] are not

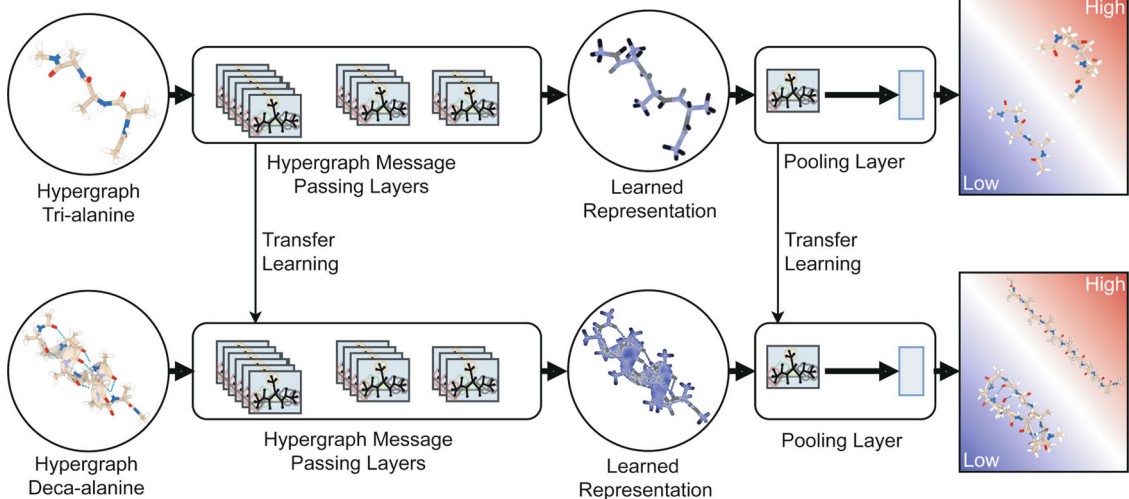

**Fig. 1 Transfer learning pipeline.** The top part of the figure represents the training of the neural network model, where the hypergraph representation of the molecules used for training (e.g., examples of the tri-alanine system) are passed through hypergraph message-passing layers to obtain hidden representations. Such representations are further processed by a pooling layer to output the probability of the input being a low free-energy conformation. The bottom part of the figure describes the transfer learning process, where the trained model is used to process examples of the target system (e.g., the deca-alanine system) and make predictions accordingly.

able to fully capture the information required to describe the potential energy (1).

Therefore, we decided to represent each molecule conformation as a hypergraph (see Methods for technical details), with vertices $V$ representing atoms and hyperedges $E = \{e_1, e_2, \ldots e_N\}$ the various types of interactions among them. In hypergraphs, each hyperedge $e$ is a set and hence it is able to describe the relation between possibly many vertices, i.e., more than two vertices. Notably, we consider $|e| = 2$ for bonds and non-bonds interactions, Coulomb and Van der Waals forces, $|e| = 3$ for angles between three atoms, and finally $|e| = 4$ for the dihedrals between planes formed by four atoms. A hyperedge feature set of size five is chosen, which stores an encoding of the type of interaction and the related feature value (e.g., the Van der Waals force). A vertex feature set of size two is chosen, which includes the mass and radius of the corresponding atom. In such a way, nodes and hyperedges of the hypergraphs are equipped with numerical features that ensure an accurate description of the interactions between atoms in each conformation assumed by the system.

Once defined an accurate representation of the system, we fed the HNN with conformations of one system (the simplest one) each labeled with a free-energy value computed through metadynamics calculations. In particular, we consider a system's conformation described by a set of coordinates $x$ and a user-defined metadynamics bias $V(x)$ as a function of a limited number of collective variables $s(x)$, which are functions of the coordinates $x$ (see refs. [44–49] and related Supplementary Note 1 for details). The free-energy of the system $F(s)$ as a function of $s(x)$ can be computed as:

$$F(s) = -\frac{1}{\beta}\ln\left(\int dx \exp(-\beta V(x))\delta(s - s(x))\right), \quad (2)$$

where $\beta$ is the inverse of the product of the Boltzmann constant $k_b$ and the temperature $T$ of the system, while $\delta(\cdot)$ is the Dirac delta function. The so computed free-energy ($F(s)$) allows for identifying the lowest energy, hence the most probable conformations of a system.

As a result, our model embeds the potential energy (1) in the proposed hypergraph representation of molecules, and predicts the free-energy (2) of a given conformation by inputting such

representation to a neural network model. More precisely, let us denote with $\mathcal{H}$ the space of all hypergraphs representing all possible molecular conformations of the system under analysis, and let us denote with $\mathcal{F}$ the space representing the free-energy values (typically $\mathcal{F} = \mathbb{R}$ is the real line). The neural network can be described as a non-linear and parametric function $g : \mathcal{H} \to \mathcal{F}$ that outputs a free-energy value $f \in \mathcal{F}$ given an input $h \in \mathcal{H}$, i.e., $g(h) = f$. As mentioned before, the neural network is trained with free-energy data obtained from metadynamics simulations.

In the transfer learning setting taken into account here, both the molecule representation and the neural network need to manage two differently sized molecular systems. Hypergraphs naturally account for this aspect by considering a variable number of vertices and hyperedges. However, neural network models capable to make global predictions on variable-size hypergraphs are not available in the literature. Therefore, we designed a novel message-passing layer that can process hypergraph-structured data of variable size, and a novel pooling layer to aggregate the information of variable-size conformations (see Methods for details).

The proposed methodology was tested on molecular systems of different complexity, the alanine, tri-alanine, and deca-alanine systems, and the results are described in the following sections.

**From alanine dipeptide to tri-alanine.** In the first experiment, we perform transfer learning from alanine dipeptide to tri-alanine. Alanine dipeptide is a relatively simple molecule, used as a reference system for conformational free-energy calculations[50–52]. In particular, it is well-known that the backbone dihedral angles $\phi$ and $\psi$ are the most relevant degrees of freedom and as such, they can distinguish the different conformations assumed by the system. In order to test the transfer learning ability of our model from alanine dipeptide to a more complex and biologically relevant system, we decided to study tri-alanine. In fact, tri-alanine represents a natural evolution of alanine dipeptide, however, it increases the complexity of the system with four additional dihedral angles. Although the structure is not long enough to fold in organized secondary structures (i.e., hairpin or helix), the number of possible conformations is considerably higher than alanine dipeptide. These conformations can

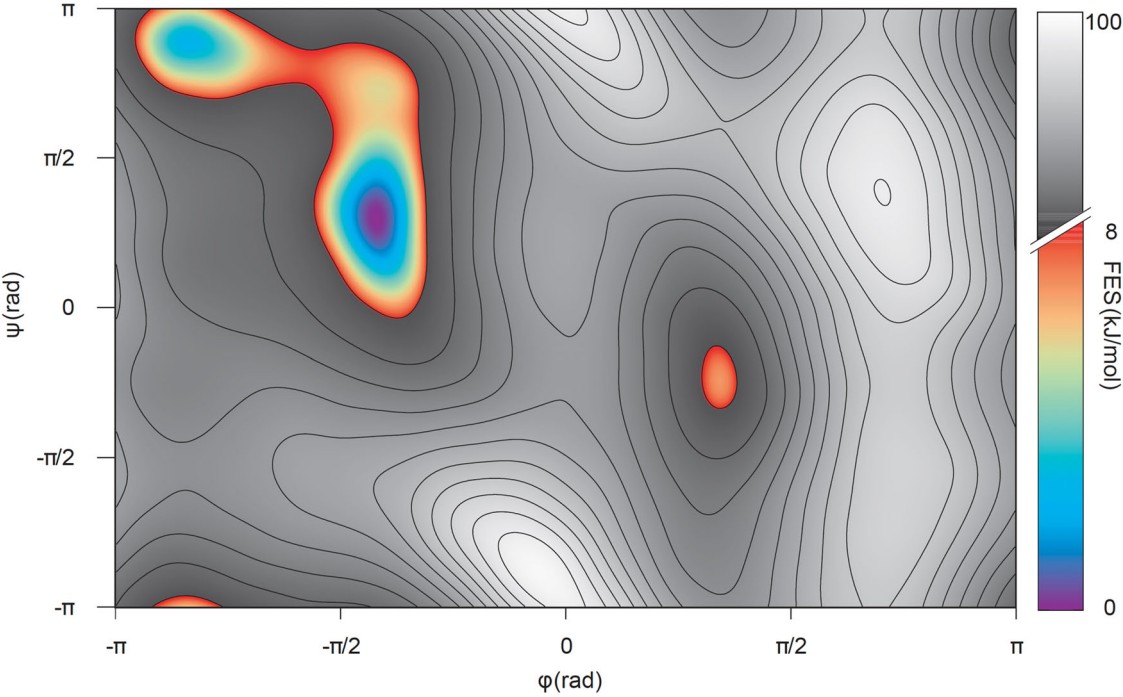

**Fig. 2 Free-energy surface for the conformational space of alanine dipeptide.** In the color range the interval from 0 to 8 kJ/mol is employed for the threshold. The rest of the surface represents high-energy conformations that have been grayed out.

be distinguished based on the combination of $\phi$ and $\psi$ angles of each residue that, taken singularly, closely reproduce the behavior seen in alanine dipeptide. For this reason, it should be feasible to train a neural network on the simpler system, trying to predict characteristics that are also relevant for the more complex one.

The structural and energetic data for alanine dipeptide were obtained from 100 ns metadynamics calculations in a vacuum, using $\phi$ and $\psi$ as collective variables (more information in the Methods section). The relative simplicity of the system allowed us to reach convergence of the free-energy calculation, thus providing a reliable ground truth for the HNN model. Instead, the tri-alanine system required 400 ns of metadynamics simulation with a more complex setup that is described in the Methods section. The conformations and the free-energy data generated by metadynamics represent the input of the HNN model, which is formed by two consecutive layers of message passing followed by a pooling layer, and a single linear layer that outputs the probability of the input being a low free-energy conformation. Supplementary Note 4 describes the model in more detail. In particular, the HNN model was trained on the alanine structural and free-energy data, and then the HNN model was transferred to the tri-alanine dataset without using any free-energy information related to the tri-alanine system, i.e., the model was trained in a zero-shot fashion. The available data is split into training, validation, and test datasets. For any five consecutive conformations in the alanine dataset, the first one was selected for training (20%), the second and third for validation (40%), and the fourth and fifth for testing (40%).

As said before, we are interested in evaluating whether HNN can distinguish between high and low free-energy conformations of the tri-alanine system. To this end, we set a threshold value equal to 8 kJ/mol for differentiating between high and low free-energy conformations. This value was chosen considering that all structures comprised in the 0–8 kJ/mol interval, where 0 kJ/mol is attributed to the global minimum, belong to known low energy and metastable states for alanine dipeptide, whereas values greater than 8 kJ/mol correspond to high-energy conformations (Fig. 2).

HNN outputs a probability value $p \in [0, 1]$ that the input denotes a low-energy conformation of the target system. This output is converted into a deterministic decision by setting a threshold $t \in [0, 1]$: $p \geq t$ indicates membership to the low-energy class; conversely, $p < t$ indicates membership to the high-energy class. In order to systematically evaluate the performance of our HNN model, and make the performance evaluation not dependent on the choice of threshold $t$, we performed the receiver operating curve (ROC) analysis[53]. The resulting ROC curve is shown in Fig. 3, denoting a relatively high AUC value of 0.89.

**From tri-alanine to deca-alanine.** The second, biologically more relevant case study considers transfer learning from tri-alanine to deca-alanine. In fact, among the poly-alanine peptides, deca-alanine represents a challenging molecule since it is able to assume secondary structures, characterized by specific alanine conformations that are not represented in alanine dipeptide and tri-alanine systems. The significantly higher structural complexity also increases the difficulty of predicting the free-energy. In Fig. 4, we report a selection of possible structures assumed by deca-alanine in a vacuum.

The deca-alanine system has been employed as a reference model by several groups in order to rank energetics in peptide folding and to test new sampling methods[54–58]. Previous works agree in reporting as the energetically preferred state is the helical conformation, passing to higher energy conformations from $\alpha$-helix, to $\pi$-helix, and finally to random coil for the unfolding state[54]. Alternative structures can also be found (i.e., $\beta$ hairpin), showing proportional or even lower free-energy estimates with respect to the helixes family[57], thus making deca-alanine a real case study of practical importance.

The deca-alanine system was simulated in a vacuum for around 700 ns. Similarly to what has been done in the tri-alanine case, the sampling of all possible secondary structures was obtained by enhancing the sampling through metadynamics, using the Root Mean-Squared Deviation (RMSD) of the $C_\alpha$ atoms

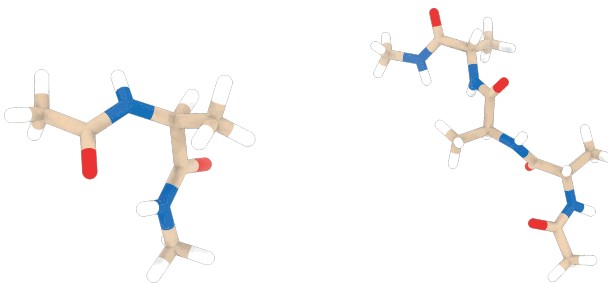
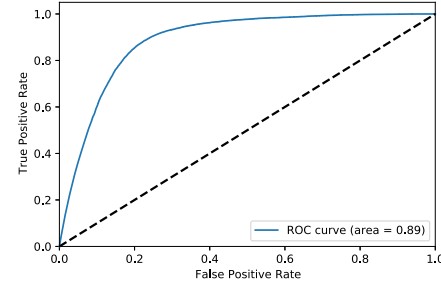

(a) A sample conformation of alanine dipeptide.

(b) A sample conformation of tri-alanine.

(c) ROC and AUC (0.89) of the HNN model trained on alanine dipeptide and tested on tri-alanine.

**Fig. 3 Molecules used in training and testing of the HNN model, along with its ROC and AUC.** Alanine dipeptide (**A**) was used in training the system, tri-alanine (**B**) was used in testing the system, and the resulting ROC and AUC (**C**).

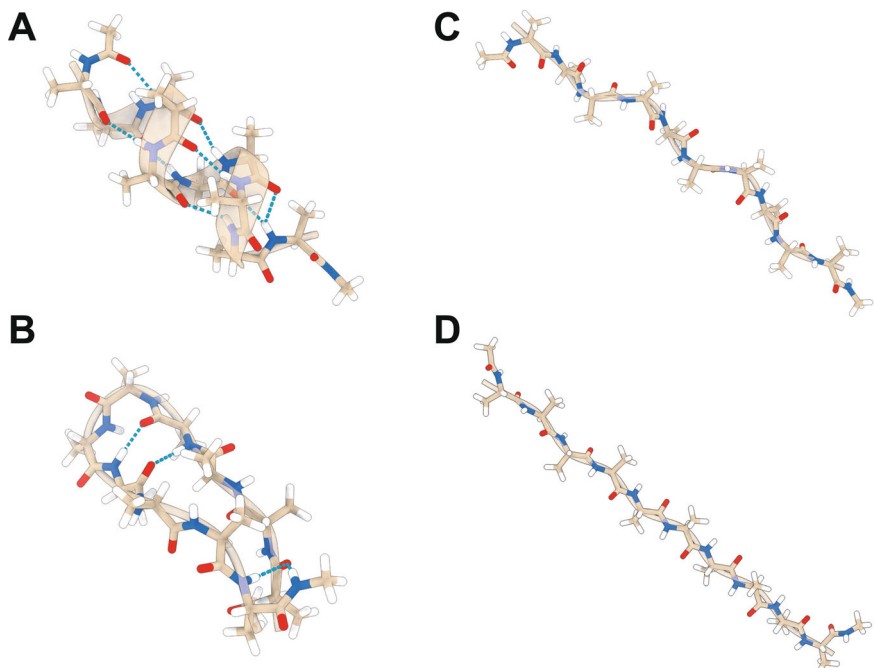

**Fig. 4 Various conformations of deca-alanine.** Deca-alanine in $\alpha$-helix (**A**), hairpin (**B**), poly-proline (**C**), and unfolded (**D**) conformation. Atoms are colored by element (i.e., hydrogen in white, carbon in tan, nitrogen in blue, and oxygen in red), the secondary structure is represented by transparent ribbons, and H bonds are shown in cyan, whenever they are relevant.

of each alanine residue as the collective variable (CV). We note that in this case, we resorted to metadynamics merely to generate very different conformations of the system by enhancing the sampling of the phase space, while we were not interested in computing the free-energy (see refs. [46,47] and related Supplementary Note 2 and Fig. S.1). To this end, more sophisticated simulation settings[59] might be used to take into account the most relevant slow degrees of freedom of the system, though requiring a long and non-trivial procedure. We note that both deca-alanine and tri-alanine are made by the same building block (i.e., alanine). However, the behavior of the deca-alanine system is completely different with respect to that of tri-alanine, as the former is able to engage intra-molecular interactions that stabilize specific secondary structures.

*Classification of low and high free-energy conformations.* Here, we assessed the feasibility of training our HNN on tri-alanine structures and the corresponding free-energy data, transferring the

acquired knowledge to classify the deca-alanine conformations as low/high free-energy conformations, again in a zero-shot fashion. We stress that no free-energy estimate of the deca-alanine system was used as supervised information for training the HNN model. This classification problem is more difficult than it might seem. In fact, it is interesting to consider that the poly-proline structure, seen as a minimum for tri-alanine, should be rather disfavored in deca-alanine, which instead prefers assuming conformations stabilized by intra-molecular H bond interactions (Fig. 5).

This kind of interaction is indeed present in any helix and $\beta$-sheet secondary structure. One single H bond typically brings a weak energetic contribution (0.5–6 kcal/mol)[60]. However, the formation of more H bonds in a molecule can stabilize even higher-order conformations, where the gain in enthalpy, thanks to the formation of such interactions, is significantly higher than the loss in entropy due to a more constrained conformation assumed by the system. As a result, the formation of helices is possible only in peptides made by a relatively high number of

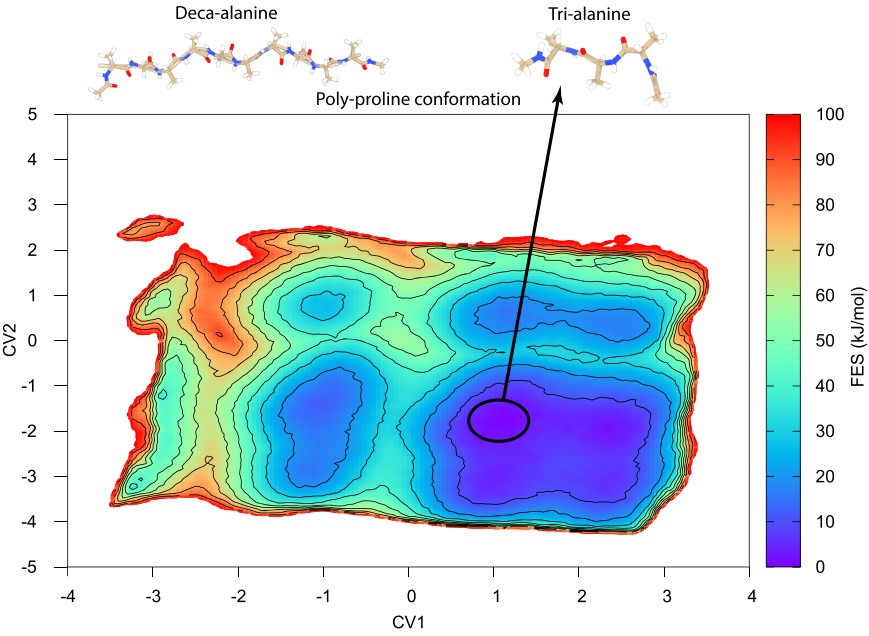

**Fig. 5 Free-energy surface for tri-alanine.** The absolute minimum in the plot corresponds to the poly-proline-like conformation, which is not the preferred conformation in the case of deca-alanine, represented in the upper left corner.

**Table 1 Classification results for deca-alanine with different thresholds.**

| Threshold | Precision ($\frac{tp}{tp+fp}$) | Recall ($\frac{tp}{tp+fn}$) |
|---|---|---|
| Low ≤ 0.45 | 0.956 | 0.836 |
| High > 0.45 | 0.979 | 0.624 |
| Low ≤ 0.5 | 0.923 | 0.950 |
| High > 0.5 | 0.921 | 0.880 |
| Low ≤ 0.55 | 0.907 | 0.999 |
| High > 0.55 | 0.877 | 0.894 |

amino acids where a number of H bonds can be engaged. Importantly, no intra-molecular H bond is observed in the training set, thus further challenging the HNN model.

As in the previous experiment, the HNN model consists of two layers of message passing for hypergraphs, followed by a pooling layer, and finally passing the resulting internal representation through a single linear layer that outputs the probability that the input represents a low free-energy conformation. This model is trained on the tri-alanine dataset containing 100,000 examples; the split considers 20% randomly chosen training data, 20% for validation, and 60% for test data.

The performance of the model over the deca-alanine system varies depending on the chosen threshold for discriminating low and high free-energy conformations. Three different representative values have been selected and the results are shown in Table 1. As in the previous experiment, to provide a more robust measure of classification performance that does not depend on the choice of a specific threshold, we performed ROC analysis and computed the AUC of our HNN model. Results are shown in Fig. 6, which denote a remarkable AUC of 0.92, thus confirming the ability of our HNN model to distinguish between high and low free-energy deca-alanine conformations with a remarkable performance.

*Secondary structure recognition.* Obtaining a converged free-energy calculation and the identification of low free-energy states as ground truth for deca-alanine is not trivial, like for many other complex molecular systems. For this reason, using only the structures generated by the simulations we challenged the HNN model in recognizing different secondary structures in an unsupervised way. More precisely, we used the HNN model to make predictions on the deca-alanine free-energy values and used those predictions to cluster conformations on the sole base of their numerical similarity. Detailed methodological aspects are discussed in Section Unsupervised secondary structure recognition.

The deca-alanine conformations generated by the atomistic simulations can be clustered in ten conformational families based on the RMSD of the alanine backbone atoms. Figure 7 shows the representation of the ten clusters together with the distribution of their $\phi$ and $\psi$ angles in a 3D Ramachandran plot. It is important to note that the HNN model does not use RMSD-based clustering information. Then, additional structures were generated from the ten most populated clusters by means of standard MD simulations. In particular, each cluster representative has been simulated with a constraint on the RMSD of the backbone atoms to produce 1000 additional structures for each cluster representative, reaching a total of 10,000 structures. More details are discussed in Supplementary Note 3.

The ten different clusters, with numerical identifiers going from 0 to 9, can be grouped into three distinct families, whose members share common structural features that should be recognized by our neural network during transfer learning:

- Helix family: clusters 1, 2, 4, and 9;
- Hairpin-like family: clusters 5 and 6;
- Extended family: all unfolded conformations (i.e., poly-proline and fully extended $\beta$ structures) in clusters 0, 3, 7, and 8.

Figure 8 shows a color map of the outcome of the statistical tests performed to assess the similarity between the distributions underlying the free-energy predictions made by the HNN model for the structures in the various clusters. Green and blue cells denote outcomes that are in agreement with our initial assumptions of energetically dissimilar and similar structures, respectively. On the other hand, yellow and red cells indicate

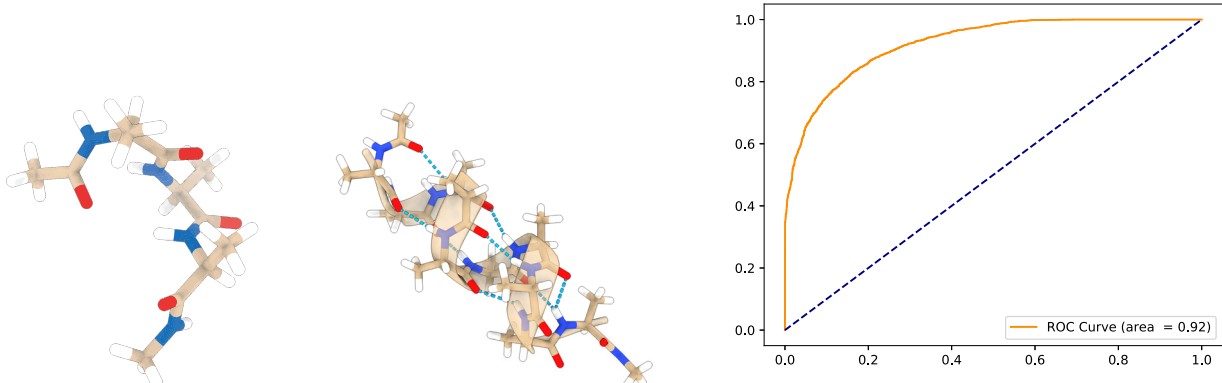

(a) A sample conformation of tri-alanine.

(b) A sample conformation of deca-alanine.

(c) ROC and AUC (0.92) for classification of deca-alanine system.

**Fig. 6 Molecules used in training and testing of the model along with its ROC and AUC.** Tri-alanine (**A**) was used in training the system, deca-alanine (**B**) was used in testing the system, and the resulting ROC and AUC (**C**).

## Helix

### Cluster 1

### Cluster 2

### Cluster 4

### Cluster 9

## Extended

### Cluster 0

### Cluster 3

### Cluster 7

## Hairpin

### Cluster 5

### Cluster 6

### Cluster 8

**Fig. 7 Ramachandran plot for a distribution of 1000 structures per cluster, which have been divided with respect to their family.** A structural representation is also offered to show the different conformations of deca-alanine.

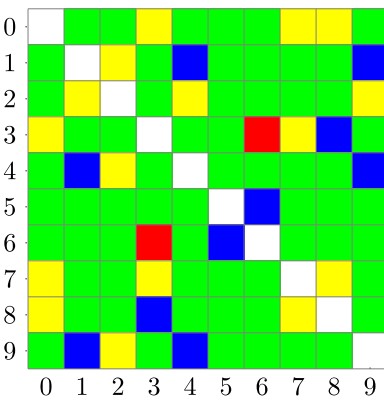

**Fig. 8 Color map of _p_ values assessing whether two clusters are in significant agreement in terms of free-energy predictions.** Green cells indicate that _p_ value is lower than the threshold (0.01) as expected, blue indicates that _p_ value was greater than the threshold as expected, yellow indicates that _p_ value was unexpectedly lower than the threshold, and red indicates that _p_ value was unexpectedly higher than the threshold. The exact values could be seen in Supplementary Note 5.

unexpected energetically dissimilar and similar structures, respectively. In detail, green cells indicate that the _p_ value is lower than the threshold (0.01) as expected, blue indicates that the _p_ value was greater than the threshold as expected, yellow indicates that _p_ value was unexpectedly lower than the threshold, and red indicates that _p_ value was unexpectedly higher than the threshold. See Methods for technical details on the statistical tests used to assess the differences.

Interestingly, the HNN model correctly recognizes structures of diverse clusters that share similar conformational properties, despite some exceptions that are reported in Table 2. Among these, the most interesting cases are discussed in the following. For example, cluster 1, which represents structures with a perfectly folded _α_-helix, is correctly recognized as a low-energy conformation, similar to the structures in clusters 4 and 9, but not with respect to cluster 2. The latter is characterized by a helical conformation similar to clusters 1, 4, and 9. However, its low _p_ values relative to the other clusters indicate that cluster 2 is energetically different from the others. A closer visual inspection of clusters 2 and 4 representative conformations, which are structurally similar, reveals that cluster 2 has the last three residues at C-terminus in a rather unfolded conformation with respect to cluster 4, which has instead an unfolded N-terminus end (see Fig. S2). Such a minor structural diversity leads to a difference in free-energy that is predicted by the HNN model. Similarly, the HNN model is able to distinguish between clusters 2 and 9, which have minor structural differences from those reported for clusters 2 and 4. On the other hand, clusters 9 and 4, which show a similar secondary structure with the same number of unfolded residues at the same end, are indicated as energetically close to the model.

Another interesting example is cluster 0 with respect to the clusters of the hairpin-like family, i.e., clusters 5 and 6. The latter is characterized by two _β_-sheets organized in an antiparallel fashion that maximizes the number of intra-molecular H bonds. On the other hand, cluster 0 is a fully extended _β_ structure, with no inter-strand interaction. Despite the similar torsion angles assumed by the alanine residues in 0, 5, and 6, HNN was able to correctly predict the diversity of 0 with respect to 5 and 6, however detecting the similarity between 5 and 6 (see Table 2).

Overall, our HNN correctly predicts most of the energetically and structurally similar conformations, however presenting a

number of outliers (e.g., clusters 0 with 3, 0 with 8, 2 with 4, 2 with 9, see Table 2 for a comprehensive list). Interestingly, for some of them, HNN seems sensitive to subtle structural differences between clusters, which would otherwise be considered similar by standard clustering methods such as those based on root mean square deviation (RMSD).

## Discussion

Transfer learning provides a framework that allows making predictions on problems with limited available data starting from different, yet related datasets. Such a framework has been extensively used in various applications, however, so far in computational chemistry it has found few applications, mostly to approximate quantum-mechanical calculations or infers material and molecular properties[31,61,62]. On the other hand, assessing free-energy values in conformational sampling by means of machine learning is, to the best of our knowledge, a novel and intriguing field of research that has recently seen more and more interest in the scientific community[9,59].

In this paper, structural features of molecules are paired with free-energy estimates of a known molecular system in order to distinguish between high or low free-energy conformations of a target system whose free-energy surface is not known, and hence not used during training. This is accomplished by means of transfer learning, which allows to exploit the information gathered on a dataset to make predictions on a different one. The proposed methodology can be of great use since it would completely replace lengthy and expensive simulations, being substituted by a machine learning model that, once trained, can output free-energy estimates in a fraction of the time. The proposed methodology, dubbed HNN in the paper, consists of two ingredients: (i) a novel hypergraph-based representation of molecules and (ii) a novel neural network model that can process hypergraphs as inputs and make decisions accordingly. More specifically, in this work we focused on a classification problem, aimed at classifying conformations into two classes, denoting high and low free-energy values. The proposed hypergraph representation allows us to fully encode multi-atom interactions of a molecular system, since it describes the interactions between two, three, and four atoms. This innovative representation goes beyond well-known graph-based representations of molecules, that are limited to modeling pairwise interactions only. The free-energy is then estimated for the target system using structural and free-energy data describing the smaller system through non-linear, black-box processing of information by means of the proposed neural network. In this respect, our work represents one of the first of its kind with the use of hypergraphs for representing the chemico-physical properties of a given molecule, thus marking a significant advance in the field of machine learning and molecular simulations.

As a first case study, we considered the problem of classifying tri-alanine conformations starting from the information gathered from the smallest possible building block, i.e. alanine dipeptide. This first test was done to assess the capability of the proposed method in a controlled setting. In fact, the tri-alanine molecule is not big enough to populate organized secondary structures such as helices and _β_-sheets, and its three-dimensional structure can be seen as a combination of three different alanine dipeptides. The obtained results showed the ability of HNN to classify the tri-alanine conformations with a remarkably high AUC value. Then, we moved to a more realistic molecular system, i.e., the deca-alanine system. This experiment was considerably more challenging since the conformational properties of deca-alanine are significantly different from the data used during training, i.e., torsion angle values of the alanine backbone in low free-energy conformations are different between tri-alanine (used as the

**Table 2 Systematic analysis of peculiar results found by the HNN model.**

| First cluster | Second cluster | Expectation | Explanation |
|---|---|---|---|
| 0 | 3 | High similarity for predictions | Cluster 0 belongs to the extended family just like cluster 3, but the latter is mainly organized in poly-proline, thus introducing a significant structural difference that is detected by the HNN model |
| 0 | 7 | High similarity for predictions | The poly-proline conformation is closely related to the $\beta$-sheet, but HNN is able to discern the two secondary structures |
| 0 | 8 | High similarity for predictions | See the explanation for clusters 0 and 3 |
| 1 | 2 | High similarity for predictions | Cluster 2 is mainly organized in $\alpha$-helix, but part of it has unfolded structures, thus justifying the observed prediction differences |
| 2 | 4 | High similarity for predictions | Cluster 2 and cluster 4 are similar both in terms of structures and distribution of dihedral angles, however, they are considered different by HNN representing an outlier. See the "Secondary structure recognition" section for discussion. |
| 2 | 9 | High similarity for predictions | As for clusters 2 and 4, also 2 and 9 are structurally and energetically similar, but predicted differently by HNN. See the "Secondary structure recognition" section for discussion. |
| 3 | 6 | Low similarity for predictions | Cluster 3 has part of its structure organized as $\beta$-sheet, similarly to cluster 6 |
| 3 | 7 | High similarity for predictions | Cluster 3 does not have a perfect poly-proline structure, while cluster 7 does, thus justifying the differences in the predictions |
| 7 | 8 | High similarity for predictions | See the explanation for clusters 3 and 7 |

The column "Expectation" contains the expected outcome based on our visual comparison of the clustered structures, while the "Explanation" column gives a justification for the deviation from the expected outcome, highlighting that our model is able to detect subtle differences and similarities between structures in different clusters.

training set) and deca-alanine. Our results show that the HNN model successfully classifies low and high-energy deca-alanine conformations with a remarkable degree of confidence, as shown in the Results section.

In addition to classifying low/high free-energy conformations in a supervised setting, we considered the application of the proposed methodology in an unsupervised setting. More precisely, we considered the possibility to cluster conformations of deca-alanine by using only structural information from alanine and tri-alanine, i.e., no free-energy values are used during training

in this case. Our results show that the HNN model is able to detect small conformational changes among all the analyzed clusters and to recognize similarities between conformations that belong to different cluster families. For instance, comparing the $p$ values computed for cluster 0 with respect to all the other clusters (see Table S1), it is interesting to note that HNN predicts cluster 0—corresponding to the fully extended conformation—energetically more similar to clusters 5 and 6—forming a $\beta$-hairpin—rather than to other extended poly-proline like structures. Indeed, the $\beta$-hairpin is formed by two $\beta$-sheets in an antiparallel

orientation connected by a turn that allows maximizing the number of intra-molecular H bonds. Cluster 0 does not form a β-hairpin, but its backbone torsion angles assume values similar to those characterizing a β-sheet secondary structure that are detected by the model. In general, the HNN model performs well in identifying clusters that are otherwise poorly classified by simple geometrical descriptors like RMSD (e.g., see the similarity between clusters 3 and 6).

The potential of such a model is huge. For instance, it might use simple building blocks (i.e., amino acids) to predict low free-energy conformations of peptides, peptidoids—often employed as drugs—as well as of proteins or part of proteins not resolved by spectroscopic experiments. From this perspective, it is useful to better understand the model functionality with the aim of further improving its prediction capability. Examples are the differences predicted by the model for clusters 2–4 and 2–9, which are expected to be similar as both assume similar α-helix conformations. The minor changes in the structural organization of the α-helix between clusters 2, 4, and 9 suggest a remarkable sensitivity of the model in detecting such differences, however, a deeper rationalization of the outlier data is necessary in the near future. Our model has proven to be efficient in classifying low and high free-energy conformations in systems made by the same "building block" amino acid (alanine) in a relatively short sequence (deca-alanine). The capability of predicting free-energy values for more complex systems made by diverse secondary structures organized in tertiary structures and multiple amino acids, remains to be investigated. In this perspective, the results obtained for clusters 5, 6, and 0, where 5 and 6 form a tertiary structure not present in 0, are encouraging.

Furthermore, although the classification performance of the HNN model was satisfactory, as demonstrated by the remarkably high AUC, the performance of the model in a regression setting to predict conformations' free-energy values was not equally good (results not shown). The free-energy prediction in a regression setting is certainly a fascinating and desirable objective to pursue in the near future, since such information is particularly useful to elucidate molecular properties (e.g., to obtain an accurate description of the free-energy landscape) and design experiments accordingly[22,63].

In conclusion, our work is a proof of concept that hypergraph-based neural networks can be successfully used to predict energetic properties for molecular systems that are otherwise inaccessible through state-of-the-art molecular simulations. Our results prompt further work in this direction, notably on developing improved neural network models and hypergraph representations able to deal with even more complex, biologically relevant systems (e.g., protein–ligand complexes), marking a significant advance in the field of molecular simulations. Finally, we note that the proposed methodology could be implemented as a run-time plug-in or a post-processing tool for molecular dynamics simulations, to identify low and high free-energy conformations that could help drive the sampling of the phase space, disclosing energetic and structural properties in an affordable computational time.

## Methods

**Molecular representation in MD**. The starting data used by our methodology are structural and topological information coming from MD simulations. Therefore, in this section, we will describe the main features of a molecule from the computational chemistry viewpoint. In MD simulations, a molecule is generally defined by a coordinate file, storing the Cartesian coordinates of each atom of the system, and a topology file, containing parameters to reproduce the physical properties of atoms. For the sake of this study, we focus on the parameters that are relevant in the classification of conformational states for the various systems under study. Understanding how a peptide or a protein is organized in the three-dimensional space in a given environment is not simple. Here, we evaluate if the information extracted from simulating a simple molecule could be used to classify the conformations of a more complex structure. As for the experiments reported in the Results section, we need to introduce two structural levels of peptides:

- primary structure—the sequence containing a list of all the amino acids comprising a given peptide (e.g., ACE-ALA-ALA-ALA-NME for tri-alanine);
- secondary structure—the three-dimensional organization of all the residues in the sequence, which might give rise to well-known patterns, like helices, β-sheets, etc.

For the latter, a key role is played by the values of the dihedral angles. Given four consecutive atoms (from 1 to 4) connected by bonds (i.e., 1 is bond to 2, 2 to 3, and 3 to 4), a dihedral (torsion) angle is the angle defined by two planes made by the first three atoms (1 to 3) and the second three (2 to 4). The result could be seen as a rotation around the bond between atoms 2 and 3, as represented in Fig. 9c.

Each amino acid has two main dihedral angles running through its backbone structure that are called $\phi$ and $\psi$, and their combination can be mapped in order to assign known secondary structure motifs to a peptide, as it has been done in the Ramachandran plots of deca-alanine in Fig. 7. Dihedral angles are included in the topology file (in fact, they contribute to the potential energy, as seen in Equation (1)) and they are provided as input to the HNN model during training. It is important to note that in order to define a specific conformation of a peptide by dihedral angles, the latter are defined only by consecutive atoms that are physically connected through bonds.

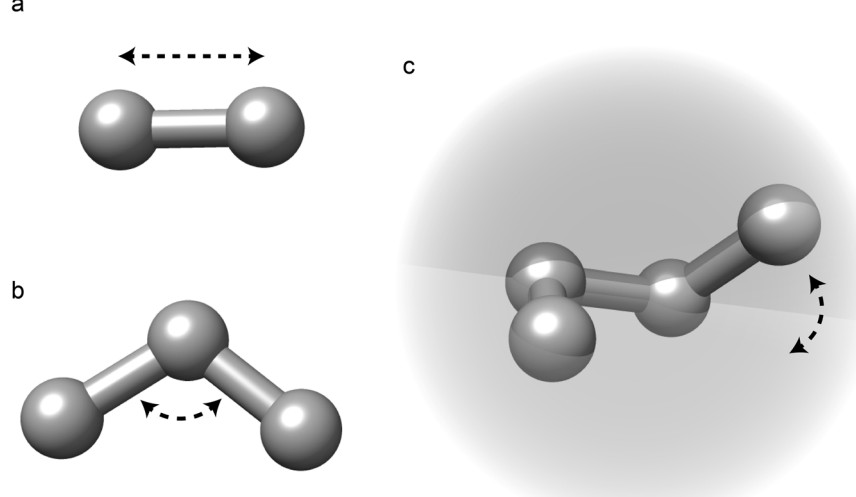

**Fig. 9 Representations of various constraints in MD simulations.** Graphical representation of the bond (**a**), angle (**b**), and dihedral angle (**c**) constraints in defining a molecule in MD simulations.

**Molecule representation as hypergraphs**. Due to the importance of higher-order interactions among atoms in describing the potential energy of conformations (1), we developed a novel hypergraph-based representation of molecules that encodes all relevant atom interactions.

Formally, a hypergraph $H(V, E, X, W)$ represents a conformation of a molecule, with $V$ being the set of all vertices (corresponding to the atoms of the molecule) and $E$ the set of hyperedges, modeling higher-order interactions, i.e. interactions between two or more vertices. Note that $E \subset \mathcal{P}(V)$, where $\mathcal{P}(V)$ is the power set of $V$, i.e. the set of all possible subsets of atoms. We consider the following four interactions: bonds and non-bonds binary relations ($|e| = 2$), angles ($|e| = 3$), and dihedrals ($|e| = 4$). $X$ is a matrix containing atom features, including atomic number and p-charge. The matrix $W \in \mathbb{R}^{|E| \times 5}$ contains features of the hyperedges. Notably, the $i$th row of the matrix $W$, $W_i$, is a vector of size five containing the following information:

$$W_i = \begin{bmatrix} 1 \text{ if } e_i \text{ is a bond, } 0 \text{ otherwise} \\ \text{Coulomb force if } e_i \text{ denotes a Coulomb interaction, } 0 \text{ otherwise} \\ \text{Van der Waals force if } e_i \text{ denotes Van der Waals interaction, } 0 \text{ otherwise} \\ \text{angle between three atoms if } |e| = 3, 0 \text{ otherwise} \\ \text{dihedral between four atoms if } |e| = 4, 0 \text{ otherwise} \end{bmatrix} \tag{3}$$

The structure of an hypergraph is represented through two matrices: a binary incident matrix $B \in \mathbb{R}^{|E| \times |V|}$

$$B_{ij} = \begin{cases} 1 & v_j \in e_i \\ 0 & \text{otherwise} \end{cases} \tag{4}$$

and an adjacency list $L \in \mathbb{R}^{|E| \times 2}$, such that $L_i = [i, j]$ indicates that $v_j \in e_i$. Both $B$ and $L$ encode the same type of information, but they are used in different ways to speed up the computations. Notably, the adjacency list is required for operations on GPU, and the incident matrix for operations running on CPU.

**Hypergraph message-passing neural network**. We design a novel message-passing neural network capable to process hypergraph-structured data. The proposed neural network model performs a series of message-passing operations on the input hypergraph followed by pooling layers to calculate a function over the whole input hypergraph. Similarly to message-passing schemes in graph neural networks[64], the use of message-passing operations allows us to significantly reduce the number of learnable parameters, which, in turn, decreases the bias and the required amount of data for training.

The proposed message-passing layer for hypergraphs employs sigmoid activation functions and performs sum aggregation. The nodes prepare a message through a linear function followed by a sigmoid activation that is sent to their hyperedges, which are then aggregated and combined with the hyperedge's features and sent back to the nodes. Finally, both the nodes and hyperedges update their internal representation. These operations are formalized as follows:

$$M_v = f_v(X_v^{(t)}) \tag{5}$$

$$W_e^{(t+1)} = g_w\left(W_e^{(t)}, \sum_{v \in e} M_v\right) \tag{6}$$

$$M_e = f_w\left(W_e^{(t)}, \sum_{v \in e} M_v\right) \tag{7}$$

$$X_v^{(t+1)} = g_v\left(X_v^{(t)}, \sum_{e \in e_v} M_e\right) \tag{8}$$

$X_v^{(t)}$ is the representation of vertex $v$ at layer $t$, $W_e^{(t)}$ is the representation of hyperedge $e$ at layer $t$, $f_v$ and $f_w$ are vertex and hyperedge messaging functions, respectively, both of which concatenate their inputs to form a vector and then apply sigmoidal functions on each element, $g_v$ and $g_w$ are vertex and hyperedge updating functions, respectively. The notation $e_v$ represents the set of hyperedges containing the vertex $v$. The updating functions apply a learnable linear transformation $L(x)$ to the current representation $x$ and add it to the incoming message $m$, i.e.,

$$g(x, m) = L(x) + m \tag{9}$$

$$L(x) = Wx + b \tag{10}$$

where $W$ and $b$ are the learnable parameters that are initialized randomly and then updated with back propagation during training. The final output is the learned representation by the current layer.

**Pooling for hypergraphs**. After the message-passing layers, a novel pooling layer is employed to produce a fixed-size, numeric representation for the input hypergraph. This is achieved by comparing the hypergraph with a set of points of interest, which, for the specific application discussed in this study, are molecular conformations that have a distinct enough internal representation after message

passing. The points of interest are selected via the K-means clustering algorithm: cluster centroids are the points of interest.

Since the conformations of different molecular systems might have very different sizes, we need to devise a mechanism that allows us to make global decisions regardless of the system size. To this end, after the points of interest are computed, for each input hypergraph, we create a fixed-size feature vector encoding the hypergraph pairwise similarity values with respect to the points of interest. The similarity degree between a hypergraph $x$ and a point of interest $p$ is computed as follows. We consider the concatenation of all vertex and hyperedge features for both the hypergraph, denoted as $\mathbf{x}$, and the point of interest, denoted as $\mathbf{p}$. We note that $\mathbf{x}$ and $\mathbf{p}$ might have different sizes, and hence direct comparisons is not possible. We, therefore, rely on a sliding window-based mechanism that assesses their similarity by considering a sliding window with size equal to the smaller structure.

Formally, the pooling layer over input $\mathbf{x}$ with points of interest $P$, with $k = |P|$ defined by the user, performs the following steps:

- For each point of interest $p_i \in P$, create a vector $\mathbf{v}_i$ containing similarity values computed with the cosine similarity between $\mathbf{x}$ and $p_i$ running over a sliding window with step 1
- Create vector $\mathbf{l} \in \mathbb{R}^{3|P|}$, and fill it in the following way:

- $\mathbf{l}_{3i} = \min(\mathbf{v}_i)$
- $\mathbf{l}_{3i+1} = \text{average}(\mathbf{v}_i)$
- $\mathbf{l}_{3i+2} = \max(\mathbf{v}_i)$

- Feed $\mathbf{l}$ to a fully connected neural network, which outputs the probability that the input conformation $x$ is a low free-energy conformation

The learnable parameters in the proposed pooling mechanism are those of the final neural network. It is, however, important to update the representations of the points of interest periodically, e.g., when the message-passing part of the network is updated.

**Scalability of the neural network operations**. The proposed molecule representation requires $3n^2 + 2n + 7e$ floating-point numbers per input conformation, where $n$ is the number of atoms and $e$ is the number of hyperedges. Each message-passing layer of the neural network requires a constant amount of space to store the weights, hence it does not depend on the size of the molecules. However, it requires $O(e + n)$ time to perform the message-passing operations, meaning that it scales linearly with respect to the size of the molecule. The pooling layer, instead, requires $k \times (e + n)$ space and $O(k \times (e + n)^2)$ time, where $k$ is the number of interest points. Assuming $k$ is much smaller than $n$ and $e$, each pooling operation scales quadratically with the molecule size.

**Transfer learning**. Transfer learning[29] is a machine learning technique used to learn models over some data distribution and transfer such models over different distributions. It is often described through its source and target distributions, as well as source and target tasks. The goal is to train a model to solve the source task on the source distribution, and then adjust it so that it can solve the target task on the target distribution.

In our experiments, we consider zero-shot transfer learning[65] between a source and a target molecular system, e.g., between alanine dipeptide and tri-alanine. Zero-shot transfer learning does not assume the availability of information about the target system during training, making it more relevant in the molecular dynamics simulation setting we are interested in. The task of interest is classification, and in particular, we are interested in classifying low and high free-energy conformations.

To ensure that the message-passing layers capture information relevant to the target system, we equip the loss function used during training with an extra regularization term in which input examples of the target system are partially processed by the network during training. Please note that such an extra regularization term does not take into account any supervised information we might have about the target system (i.e., its free-energy), but only structural information. To this end, we calculate a representative structure of the target system, and we pass it through the message-passing layers.

Formally, for each target conformation observed during training, we construct a vector $D_i$ such that the $j$th entry is the $j$th feature of the related hypergraph. Stacking $D_i$ gives us a matrix, $D$. We calculate the principal axes of $D$ through their right singular vectors (eigenvectors of $A^T A$), and sum them, obtaining the representative $r_D$ for the target system. We note that $r_D$ represents an approximation of the variance of the target distribution.

The loss function used during training reads:

$$\text{BinaryCrossEntropyLoss} + l_2 + \text{TargetLoss} \tag{11}$$

where $l_2$ denotes the l2 penalty on the learnable weights, and the binary cross entropy loss is defined as

$$\frac{1}{|N|} \sum_{i=1}^{N} y_i \log(p(y_i)) + (1 - y_i) \log(1 - p(y_i)) \tag{12}$$

The third term refers to the aforementioned extra regularization on the target system distribution:

$$\| \text{HMPNN}_2(\text{HMPNN}_1(r_D, W_1), W_2)\|_2 \qquad (13)$$

where $\text{HMPNN}_i$ denotes the $i$th message-passing layer (without losing generality, we assume two message-passing layers, although this can be generalized to any number of layers) for hypergraphs with weights $W_i$, and $r_D$ is the representative defined as above.

**Unsupervised secondary structure recognition**. Due to the lack of ground truth for the deca-alanine free-energy landscape, we perform an additional test to validate the results of transfer learning. This test uses the trained HNN model to perform an unsupervised secondary structure recognition, assessing whether the HNN model is able to learn the secondary structures of the target system in a transfer learning setting.

We make the assumption that similar secondary structures of the target system have similar free-energy values, and that such similarities can be captured by relying only on the information of the source system used during training. To test the validity of our assumption, we collect all predicted free-energy values for the structures in the various clusters, and compare their distributions with statistical tests to check for significant differences. Notably, we used the Wilcoxon signed rank test[66] to check if the distributions underlying the prediction values are significantly different or not. If the distributions are different according to a prescribed threshold ($p < 0.01$), then we say that the HNN model predictions for the two clusters are in disagreement, i.e., they are significantly different.

## Data availability

All data that support the finding of this study have been deposited in Zenodo with the accession code (7299776) https://zenodo.org/record/7299776.
  The data used for this research is available at https://zenodo.org/record/7299776.

## Code availability

The source code for the neural networks could be found at https://github.com/MCSH/chemical_transfer_learning_hypergraph.

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

## Acknowledgements
L.L. acknowledges support from the Canada Research Chairs program. V.L. acknowledges support from the European Research Council ("CoMMBi" ERC grant agreement no. 101001784), the Swiss National Supercomputing Centre (CSCS; project ID s1150) and the Italian MIUR-PRIN 2017 (2017FJZZRC).

## Competing interests
The authors declare no competing interests.
