## [Peer Review File · Communications Chemistry]

Reviewers' comments:

Reviewer #1 (Remarks to the Author):

Heydari et. al. propose a neural network-based methodology to predict energetically favorable configurations of large biomolecules without performing computationally expensive simulations. The neural network is trained with various configurations of a smaller biomolecule where the associated free energies are known from molecular dynamics. Using a transfer learning algorithm, the network then determines whether a given configuration of the larger target molecule is a low- or high-energy configuration.

The topic of the article is certainly intriguing and could be useful for further development of machine learning-based methodologies for free-energy calculations. However, the current version of the article requires substantial improvements in terms of both presentation and scientific details. A major shortcoming in the way the article is written is the lack of discussion of the limitations and scope of improvements of the proposed methodology.

The article would be suitable for publication once the following concerns are addressed.

- Compared to the journal standard, the article is poorly written and requires heavy language editing. It contains several typos, grammatical errors, inappropriate choices of words, and possibly formatting errors with citations. A few examples follow:

- 1) "use cases", "impacted on", "very same" in the abstract.
- 2) "similarly to" in section 2.3
- 3) "unique the choice of HNN" in section 3.

- Is AUC a well-known generalized acronym? The authors are suggested to mention at least once the elaborated form of the same for the convenience of the general reader.

- There are possibly some errors with the way the citations have been formatted. The authors are suggested to recheck it. For example, in the first paragraph of the introduction on page 2 the authors write, "ML techniques have been originally applied to chemo-informatics problems - prediction of compounds properties like solubility, toxicity etc. - thanks to the relative abundance of experimental data [1, 33]". Here 33 papers are cited together, including the MD program suite AMBER18!

- The caption of Figure 8 is inconsistent with the texts explaining Figure 8, creating confusion for the reader. The authors should double-check what green and yellow cells indicate. In the text the authors write "A p-value less than 0.01 (the significance threshold value) indicates that the predictions for the two clusters are significantly different from each other; green and yellow cells indicate that the two clusters are not significantly different, whereas red and blue say otherwise." However, according to the caption, "Green cells indicate that p-value is lower than threshold (0.01) as expected" and "yellow indicates that p-value was unexpectedly lower than threshold".

- For the sake of completeness, the authors should add another entry to Table 2 explaining the unexpected outcome for the comparison between clusters 2 and 9. Why do they choose not to comment on it?

- The discussion section should contain a critical assessment of the method explaining when it would work best and when it won't. Also, the authors should be specific about how the method can be further improved and the challenges toward those improvements. For example, any guess about where the method went wrong in the context of the comparison between clusters 2 and 4?

- Section 4 (Methods), in general, should be written with more clarity and details so that an interested reader is able to implement the method and reproduce the results. Perhaps it would be better to

include a couple of diagrams explaining the hypergraph representation of the molecule and hypergraph message passing. For example, Figure 1 is too schematic, and it is hard to comprehend the details of the technicalities from it.

- In the text following Eq. (5) in Sec. 4.2, what is represented by the angular brackets? Do they represent an average or an inner product or something else?

- For Eq. (8), the authors should give the functional form of f_w since it is a sigmoid containing two arguments unlike f_v in Eq. (6).

- What are the parameters for the linear transformation $L(x)$? Are they initialized randomly and then trained to certain values?

- The authors should discuss whether there is any loss of information during message passing and, if so, how significantly they can affect the final results.

Reviewer #2 (Remarks to the Author):

This manuscript reports a new graph neural network architecture devoted to transfer learning of a classification task between molecular systems. In its current form, the manuscript suffers from significant flaws, detailed below, and the tests do not demonstrate the usefulness of the method. As it is, it appears to be an energy prediction tool that provides less information than state-of-the-art methods.

(1) The manuscript often refers to predictions of the "free energy" of "states", but in practice the neural network acts on individual configurations and is trained on purely energetic data on these configurations, aka microstates, for which there is no associated concept of free energy. All mentions of free energy are therefore unwarranted. There is an attempt to justify this on p. 4 with the argument that entropic differences are negligible for this category of molecular systems, and therefore free-energy differences can be approximated by energy differences. Not only does this argument set a particularly low bar for methods that claim to perform free energy calculations (they should just be called energy calculations), it also ignores the conceptual distinction between the potential energy of microstates and the enthalpy of thermodynamic states. Strangely, the text first states "we are interested in obtaining an estimate of the free-energy difference, which contains the potential energy E_p but is not limited to that" and immediately proceeds to dismiss any difference between those two quantities.

This is accompanied by Equation (2), which is incorrect, as it mixes up the probability density of configurations $\rho(x)$, which should not involve a density of states, and the probability density of the energy $\rho(E)$, which does. Equation (3) presents a generally relevant approximation of Gibbs free energy differences by Helmholtz free energy differences, but uses an incorrect justification for it. Combined, these represent a severe mistreatment of statistical thermodynamics.

(2) Now that it has been clarified that this approach deals with the potential energy microstates, the reference task is energy prediction, of which the classification of high- and low-energy configurations (not "states") is a less informative subtask. Therefore, the references to gauge the practical usefulness of this approach are current energy prediction methods - to the extent that they are transferable, as transfer learning is an intrinsic feature of this approach. This is the case of molecular mechanics force fields and of several machine-learned energy models, including some that are transferable to a similar degree as this approach, as far as I can judge.

The most obvious comparison is MM force fields. Indeed, the architecture itself is highly reminiscent of

that of a typical force field, with explicit bonded and non-bonded terms. This is particularly striking when considering two terms in the hyperedge features: a Coulomb force and a van der Waals force. These are not geometric features, but force field terms: although no details are given, I expect the definition of these terms to involve force field parameters (Lennard-Jones coefficients and partial charges) which are themselves transferable to various degrees. In this sense, evaluating the network itself seems to involve a partial force field calculation. So it seems necessary to evaluate the benefits of this approach with respect to a FF calculation. On the test systems given here, for which optimized force fields are available, the classification does not bring any useful information. A much more convincing test would be transfer to systems for which there are no existing force fields. And then the method would have to be appraised against the state of the art: automated force field parameterization methods, and other machine-learned energy models.

(3) Alternately, this architecture may be useful for a task unrelated to energy prediction, and it should then be tested on that task.

Reviewer #3 (Remarks to the Author):

I am really at loss with this paper. Perhaps it has novel concepts and potential to be useful, but the current version makes it very hard to assess that. Perhaps the manuscript can be improved with a substantial rewrite and I would be happy to review that again.

Point 1: The authors start by claiming their intention is to classify high and low free energy states, and to "characterize the potential energy of a conformation". I do not understand what they mean with the part in quotes.

* (A) Does this mean a binary classification of "low free energy" and "high free energy"? If so, why not just use the force-field to get potential energy and get the job done? Why is the approach here needed? Perhaps there are non-trivial entropic contributions that simply the potential energy can not capture - if so, this needs to be demonstrated numerically.

* OR, (B) Does this mean a higher resolution than binary classification? That would be equivalent to learning a coarse free energy. Please provide clear statements and results if this is the case.

Point 2: In the manuscript itself, the authors cluster various secondary structures with similar free energy and show that their model can distinguish them. This seems like a commendable task where the model does well and could be novel. However the results once again are provided in a very convoluted manner.

Perhaps it is in the combination of points 1 and 2 that the manuscript is useful. This would allow taking a sequence and clustering different secondary structures ranked as per their free energy. But this is not clear to me from this draft.

Some further comments:

1. what would be the scalability of the approach for bigger proteins in terms of number of residues?
2. what is really the input needed for this method? Is it a free energy simulation that has visited everywhere? Is it a collection of PDBs? Is it coarse-grained MD? Please clarify cleanly.
3. Finally the english quality is quite poor. In addition technical terms are used very causally - for instance, AUC in abstract.

University of Lugano USI - Switzerland
Faculty of Biomedical Sciences
Euler Institute

Prof. Dr. Vittorio Limongelli
Full Professor in Pharmacology
ERC Investigator “CoMMBi”
USI-Campus EST, via la Santa, 1
CH-6900 Lugano, Switzerland
Phone: +41 58 666 4293
e-mail: limonv@usi.ch

Lugano, 23th August

Dear Reviewers,

We are pleased that all of you found our manuscript of interest and we thank you for the comments that we have found helpful to improve the quality of the manuscript. We have largely revised the original manuscript to improve readability and address your concerns. All the changes are highlighted in red in the revised version of the manuscript. In the following we answer point-by-point all the comments you raised.

Reviewer 1

1) Compared to the journal standard, the article is poorly written and requires heavy language editing. It contains several typos, grammatical errors, inappropriate choices of words, and possibly formatting errors with citations. A few examples follow:

- 1) “use cases”, “impacted on”, “very same” in the abstract.*
- 2) “similarly to” in section 2.3*
- 3) “unique the choice of HNN” in section 3.*

We have largely revised the text and believe that the manuscript is now fluent and reads well. We also hope the Reviewer enjoys the reading.

2) Is AUC a well-known generalized acronym? The authors are suggested to mention at least once the elaborated form of the same for the convenience of the general reader.

Yes, the AUC acronym is generally well-known in Machine Learning and Data Science.

We agree with the Reviewer and we have made the acronym explicit in the abstract.

We have also added a reference on ROC and AUC, “*Fawcett. An Introduction to ROC Analysis. Pattern Recognition Letters, 27(8):861–874, June 2006*”.

3) *There are possibly some errors with the way the citations have been formatted. The authors are suggested to recheck it. For example, in the first paragraph of the introduction on page 2 the authors write, “ML techniques have been originally applied to chemo-informatics problems - prediction of compounds properties like solubility, toxicity etc. - thanks to the relative abundance of experimental data [1, 33]”. Here 33 papers are cited together, including the MD program suite AMBER18!*

We employed the standard template for references of latex where [1, 33] means that references 1 AND 33 are cited (i.e. two papers are cited), and not from 1 to 33 that would report the hyphen between numbers (1-33). However, formatting of references and the text will also be done by the editorial office once the paper is accepted for publication - this task is very easy with latex.

4) *The caption of Figure 8 is inconsistent with the texts explaining Figure 8, creating confusion for the reader. The authors should double-check what green and yellow cells indicate.*

In the text the authors write “A p-value less than 0.01 (the significance threshold value) indicates that the predictions for the two clusters are significantly different from each other; green and yellow cells indicate that the two clusters are not significantly different, whereas red and blue say otherwise.” However, according to the caption, “Green cells indicate that p-value is lower than threshold (0.01) as expected” and “yellow indicates that p-value was unexpectedly lower than threshold”.

We do apologise for the inconsistency and we have revised the part of the main text referring to Table 8. This is now in agreement with the related caption. The revised text reads:

“Figure 8 shows a colour map of the outcome of the statistical tests performed to assess the similarity between the distributions underlying the free-energy predictions made by the HNN model for the structures in the various clusters. Red and yellow cells indicate p-values that are in disagreement with our initial assumptions about the families containing similar conformational clusters, whereas green and blue cells denote outcomes that are in agreement with our initial assumptions. Notably, green cells indicate that p-value is lower than threshold (0.01) as expected, blue indicates that p-value was greater than threshold as expected, yellow indicates that p-value was unexpectedly lower than threshold, and red indicates that p-value was unexpectedly higher

than threshold. See Methods for technical details on the statistical tests used to assess the differences.”

5) For the sake of completeness, the authors should add another entry to Table 2 explaining the unexpected outcome for the comparison between clusters 2 and 9. Why do they choose not to comment on it?

We acknowledge the Reviewer’s suggestions adding cluster 2 and 9 to Table 2. The explanation of the unexpected outcome for cluster 2 and 9 is the same for cluster 2 and 4. For this reason, we omitted this explanation in the original manuscript. However, prompted by the Reviewer’s comment, we have now added a short description on these clusters in the revised text.

6) The discussion section should contain a critical assessment of the method explaining when it would work best and when it won’t. Also, the authors should be specific about how the method can be further improved and the challenges toward those improvements. For example, any guess about where the method went wrong in the context of the comparison between clusters 2 and 4?

We acknowledge the Reviewer’s request adding the analysis of the computational complexity in Section 4.5 and explaining in more detail the potential and limitations of the method. In particular, we write:

“The potential of such a model is huge. For instance, it might use simple building blocks (i.e., aminoacids) to predict low free-energy conformations of peptides, peptidoids - often employed as drugs - as well as of proteins or part of proteins not resolved by spectroscopic experiments. In this perspective, it is useful to better understand the model functionality with the aim of further improving its prediction capability. Examples are the differences predicted by the model for clusters 2-4 and 2-9, which are expected to be similar as both assume similar alpha-helix conformations. The minor changes in the structural organization of the alpha-helix between cluster 2, 4 and 9 suggest a remarkable sensitivity of the model in detecting such differences, however a deeper rationalization of the outlier data is necessary in the near future.

Our model has proven to be efficient in classifying low and high free-energy conformations in systems made by the same "building block" amino acid (alanine) in a relatively short sequence (deca-alanine). The capability of predicting free-energy values for more complex systems made

by diverse secondary structures organized in tertiary structures and multiple amino acids, remains to be investigated. In this perspective, the results obtained for clusters 5, 6 and 0, where 5 and 6 form a tertiary structure not present in 0, are encouraging.

Furthermore, although the classification performance of the HNN model was satisfactory, as demonstrated by the remarkably high AUC, the performance of the model in a regression setting to predict conformations' free-energy values was not equally good (results not shown). The free-energy prediction in a regression setting is certainly a fascinating and desirable objective to pursue in the near future, since such information is particularly useful to elucidate molecular properties (e.g. to obtain an accurate description of the free-energy landscape) and design experiments accordingly (doi: 10.1002/wcms.1455; doi: 10.3389/fmolb.2021.712085).

In conclusion, our work is a proof of concept that hypergraph-based neural networks can be successfully used to predict energetic properties for molecular systems that are otherwise inaccessible through state-of-the-art molecular simulations.

Our results prompt further work in this direction, notably on developing improved neural network models and hypergraph representations able to deal with even more complex, biologically relevant systems (e.g. protein-ligand complexes), marking a significant advance in the field of molecular simulations.

Finally, we note that the proposed methodology could be implemented as a run-time plug-in or a post-processing tool for molecular dynamics simulations, to identify low and high free-energy conformations that could help driving the sampling of the phase space, disclosing energetic and structural properties in an affordable computational time.”

7) Section 4 (Methods), in general, should be written with more clarity and details so that an interested reader is able to implement the method and reproduce the results. Perhaps it would be better to include a couple of diagrams explaining the hypergraph representation of the molecule and hypergraph message passing. For example, Figure 1 is too schematic, and it is hard to comprehend the details of the technicalities from it.

We thank the Reviewer for the comment. We will release the code for reproducing the results once the paper is accepted for publication. This will allow any other interested researcher to look at the details and also use our implementation for replicating the results. For the time being, we

make available the code for the Editor and Reviewers at https://github.com/MCSH/chemical_transfer_learning_hypergraph

We believe that Figure 1 offers a good compromise between the technical details and the conceptual steps that form the proposed methodology. This balance should allow the reader to get an initial idea of the main steps that occur when processing the data. The technical details are reported in the Methods section. This information together with the code and a README file with user instructions and a tutorial section will render the method publicly available and accessible to even inexperienced users. We did the same for funnel-metadynamics, a binding free-energy method we reported in Nature Protocols [a] which is now world-widely used.

[a] S. Raniolo and V. Limongelli. Ligand binding free-energy calculations with funnel metadynamics. *Nature Protocols*, 15:2837–2866, 2020. doi: 10.1038/s41596-020-0342-4

8) In the text following Eq. (5) in Sec. 4.2, what is represented by the angular brackets? Do they represent an average or an inner product or something else?

H is the adjacency list of the hypergraph (now renamed L). So it is just listing all vertices that belong to the hyperedges. We have changed the notation and used angular brackets instead, but their meaning remains the same. We have also clarified why there are two matrices encoding the structural information of hypergraphs.

9) For Eq. (8), the authors should give the functional form of f_w since it is a sigmoid containing two arguments unlike f_v in Eq. (6).

Both f_w and f_v operate by first concatenating their inputs (regardless of how many they might be) into a single vector and then applying a sigmoid function element-wise to such a vector, finally producing another vector as output. We modified the text accordingly.

We stress again that, once the paper is accepted, we will publicly release the code of the neural network and the other relevant components.

10) What are the parameters for the linear transformation $L(x)$? Are they initialized randomly and then trained to certain values?

The parameters of the linear transformation are a weights matrix and a bias vector, both of which are randomly initiated and then updated by means of back propagation during training (together with all other parameters of the neural network model). We added a description and the formula for linear transformation to the text.

11) The authors should discuss whether there is any loss of information during message passing and, if so, how significantly they can affect the final results.

We agree with the Reviewer that characterizing the “loss of information” would be very informative. Unfortunately, it is not trivial to define such a concept in graph neural networks, especially considering the fact that we are working with data described as hypergraphs. A thorough discussion is thus beyond the scope of this paper. Nonetheless, we refer the reader to “*What graph neural networks cannot learn: depth vs width*, Loukas A. *International Conference on Learning Representations, 2020*” where the author discusses the limitation of message passing in graph neural networks, i.e. neural networks processing graph-structured data. The main result of that paper states that, under suitable conditions of depth and width of the network, graph neural networks based on message passing are Turing universal.

However, we stress once more that we deal with hypergraphs. Therefore, it is not obvious if what Loukas reported also applies to our case.

Reviewer 2

1) The manuscript often refers to predictions of the "free energy" of "states", but in practice the neural network acts on individual configurations and is trained on purely energetic data on these configurations, aka microstates, for which there is no associated concept of free energy. All mentions of free energy are therefore unwarranted.

We thank the Reviewer for raising this point that allows us to describe in more detail the properties of our hypergraph neural network. As correctly written by the Reviewer, our model acts on individual configurations and not on states. We note that the correct terminology is conformation (as we use in the original version of the manuscript) instead of configuration since in conformers the stereochemistry of the atoms forming the molecule does not change while it

does in configurations. Therefore, we have replaced the word “state” with “conformation” throughout the text where appropriate.

Nonetheless, we would like to clarify two things. First, each conformation is represented as a hypergraph. This is done in order to capture the interaction between 2, 3, and 4 atoms, which are fundamental to properly describe the potential energy of a conformation. Second, each conformation is processed as an input of the neural network model, whose output is the free-energy of the conformation. To this end, the neural network is trained on free-energy data (not energy or potential energy) obtained from metadynamics calculations (for theory please see refs. doi: 10.1073/pnas.202427399; doi: 10.1103/PhysRevLett.100.020603). This information is used as a target function by the neural network that, through a series of non-linear functions implemented by the described convolution and pooling operations, outputs free-energy predictions for a given input conformation of the system under study.

This concept is now clearly reported in the revised manuscript, and in particular in the section entitled “Molecular representation and processing”.

2) There is an attempt to justify this on p. 4 with the argument that entropic differences are negligible for this category of molecular systems, and therefore free-energy differences can be approximated by energy differences. Not only does this argument set a particularly low bar for methods that claim to perform free energy calculations (they should just be called energy calculations), it also ignores the conceptual distinction between the potential energy of microstates and the enthalpy of thermodynamic states. Strangely, the text first states "we are interested in obtaining an estimate of the free-energy difference, which contains the potential energy E_p but is not limited to that" and immediately proceeds to dismiss any difference between those two quantities.

We would like to stress that we did not ignore any conceptual distinction. As explained in the previous point, our neural network computes free-energy and not energy. Some energetic terms, i.e. Eq. 1 with bond, non-bond, angles, dihedrals, typically considered in the potential energy master equation of a general atomistic force field, are encoded in the proposed hypergraph representation of conformations.

3) This is accompanied by Equation (2), which is incorrect, as it mixes up the probability density of configurations $\rho(x)$, which should not involve a density of states, and the probability density of the energy $\rho(E)$, which does. Equation (3) presents a generally relevant approximation of Gibbs free energy differences by Helmholtz free energy differences, but uses an incorrect justification for it. Combined, these represent a severe mistreatment of statistical thermodynamics.

We thank the Reviewer for pointing this out. We have revised Eq. 2 that now refers to free energy as a function of conformations, and removed Eq. 3, which was confusing for the reader as no potential or internal energy was computed by the neural network.

4) Now that it has been clarified that this approach deals with the potential energy microstates, the reference task is energy prediction, of which the classification of high- and low-energy configurations (not "states") is a less informative subtask. Therefore, the references to gauge the practical usefulness of this approach are current energy prediction methods - to the extent that they are transferable, as transfer learning is an intrinsic feature of this approach. This is the case of molecular mechanics force fields and of several machine-learned energy models, including some that are transferable to a similar degree as this approach, as far as I can judge.

The most obvious comparison is MM force fields. Indeed, the architecture itself is highly reminiscent of that of a typical force field, with explicit bonded and non-bonded terms. This is particularly striking when considering two terms in the hyperedge features: a Coulomb force and a van der Waals force. These are not geometric features, but force field terms: although no details are given, I expect the definition of these terms to involve force field parameters (Lennard-Jones coefficients and partial charges) which are themselves transferable to various degrees. In this sense, evaluating the network itself seems to involve a partial force field calculation. So it seems necessary to evaluate the benefits of this approach with respect to a FF calculation. On the test systems given here, for which optimized force fields are available, the classification does not bring any useful information. A much more convincing test would be transfer to systems for which there are no existing

force fields. And then the method would have to be appraised against the state of the art: automated force field parameterization methods, and other machine-learned energy models.

We have clarified in point 1 that our neural network computes free energy and not potential energy. As a consequence, it should be clear the usefulness of our model, that is, predicting free-energy of conformations for systems otherwise hardly investigated due to the large number of degrees of freedom that hamper the free-energy calculation convergence e.g. via metadynamics used in the case of deca-alanine in our work. In fact, the main motivation behind our work is that obtaining free energy predictions for large systems - by means of metadynamics or other methods - is very challenging not to say unfeasible. Our goal is to exploit the information obtained from simpler systems and transfer such knowledge to more complex systems.

That said, prompted by the Reviewer's comment, we have performed the experiment of clustering diverse secondary structures of deca-alanine, as discussed in Section 2.3.2, using the potential energies obtained from simulations with the Amber force field. The results are remarkably different with respect to those provided by our neural network trained with free-energy data (whose results were already discussed in Section 2.3.2 of the original manuscript). The results are reported in the following figures:

The predictions made using the potential energy (left panel) are not able to cluster structures in an accurate way. This can be easily recognized by the presence of a large number of red cells. Note that we used the same color coding of the paper, which we report in the following for sake of simplicity.

Green cells indicate that p-value is lower than threshold 0.01 (i.e., conformations are dissimilar based on free-energy prediction) as expected, blue indicates that p-value was greater than threshold (i.e., conformations are similar based on free-energy prediction) as expected, yellow indicates that p-value was unexpectedly lower than threshold, and red indicates that p-value was unexpectedly higher than threshold.

On the right panel, we report the results obtained with the neural network trained on free-energy data, already shown in the paper.

This test demonstrates the utility of our model and the necessity of integrating free-energy data into the neural network to obtain accurate grouping of secondary structures of deca-alanine.

We did not add this additional test in the revised manuscript because, in our opinion, it is out of the scope of our work. However, we are willing to include such results upon the Reviewer's request.

5. Alternately, this architecture may be useful for a task unrelated to energy prediction, and it should then be tested on that task.

We agree with the Reviewer that there are other potential applications of the proposed neural network model that being a general method could be exploited to transfer-learn other properties between molecular systems. More investigations in this direction will be arguments of future investigations.

Reviewer 3

1) I am really at loss with this paper. Perhaps it has novel concepts and potential to be useful, but the current version makes it very hard to assess that. Perhaps the manuscript can be improved with a substantial rewrite and I would be happy to review that again.

Point 1: The authors start by claiming their intention is to classify high and low free energy states, and to "characterize the potential energy of a conformation". I do not understand what

they mean with the part in quotes.

Does this mean a binary classification of "low free energy" and "high free energy"? If so, why not just use the force-field to get potential energy and get the job done? Why is the approach here needed? Perhaps there are non-trivial entropic contributions that simply the potential energy can not capture - if so, this needs to be demonstrated numerically.

We have acknowledged the Reviewer's request largely revising the manuscript to improve its readability and clarity.

We have also followed the Reviewer's comment discussing in more detail the property of our hypergraph neural network. The proposed neural network model predicts free-energy values for individual conformations of a system that is much more complex (i.e., endowed with a higher number of degrees of freedom and different entropy) than the one used for training the neural network model. In fact, the main motivation behind our work is that computing free-energy for large systems by means of metadynamics or other simulation methods is very challenging and time-consuming. Therefore, we propose a methodology to exploit the knowledge (in terms of free energy) that we have for simpler systems (e.g. alanine) to predict the free energy of larger systems (e.g. deca-alanine). In our paper, we focus on the specific task of classifying conformations in two classes, describing conformations with high and low free-energy values, respectively. This is obtained by means of two novel technical components: (1) a novel hypergraph representation for conformations (one conformation is represented by one hypergraph), and (2) a novel neural network model that takes such hypergraphs as inputs and outputs the associated free-energy predictions.

Such aspects are now clearly described in the revised manuscript.

Finally, as suggested by the Reviewer, we have performed additional calculations to provide a numerical demonstration of the utility of our hypergraph neural network model. In detail, we have carried out the experiment of clustering diverse secondary structures of deca-alanine, as discussed in Section 2.3.2, using the potential energies obtained from simulations with the Amber force field. The results are remarkably different with respect to those provided by our neural network trained with free-energy data (whose results were already discussed in Section 2.3.2 of the original manuscript). The results are reported in the following figures:

The predictions made using the potential energy (left panel) are not able to cluster structures in an accurate way. This can be easily recognized by the presence of a large number of red cells. Note that we used the same color coding of the paper, which we report in the following for sake of simplicity.

Green cells indicate that p-value is lower than threshold 0.01 (i.e., conformations are dissimilar based on free-energy prediction) as expected, blue indicates that p-value was greater than threshold (i.e., conformations are similar based on free-energy prediction) as expected, yellow indicates that p-value was unexpectedly lower than threshold, and red indicates that p-value was unexpectedly higher than threshold.

On the right panel, we report the results obtained with the neural network trained on free-energy data, already shown in the paper.

This test demonstrates the utility of our model and the necessity of integrating free-energy data into the neural network to obtain accurate grouping of secondary structures of deca-alanine.

We did not add this additional test in the revised manuscript because, in our opinion, it is out of the scope of our work. However, we are willing to include such results upon the Reviewer's request.

2) In the manuscript itself, the authors cluster various secondary structures with similar free energy and show that their model can distinguish them. This seems like a commendable task where the model does well and could be novel. However the results once again are provided in a very convoluted manner.

We have revised the entire section to improve readability and clarity of the writing.

3) Perhaps it is in the combination of points 1 and 2 that the manuscript is useful. This would allow taking a sequence and clustering different secondary structures ranked as per their free energy. But this is not clear to me from this draft.

We specially thank the Reviewer for raising this point and yes, this could be another useful application of the methodology. In fact, what the Reviewer is proposing is very similar to the unsupervised secondary structure recognition we discussed in our work.

The first application of the proposed methodology consists of showing the capability of the neural network model to classify low and high free-energy conformations of molecular systems in a transfer learning setting, i.e. systems whose free-energy landscape is not characterized. This is possible thanks to (i) the hypergraph-based representation, which allows to process molecular systems of very different sizes and to include features describing multi-atom interactions (like dihedrals), and (ii) the transfer learning setting used for training and making predictions with the neural network. This application makes use of supervised information related to the free energy of simple systems used during training.

The second application we discussed in the paper shows how the proposed methodology (representation and neural network) can be used in an unsupervised setting to group together diverse secondary structures with similar physico-chemical properties (e.g., number and types of H-bonds) based on free-energy predictions. Again, also this application operates in a similar transfer learning setting, i.e. we use information from simple systems (alanine and tri-alanine) during training and the testing phase is performed on a more complex system, i.e. deca-alanine. Prompted by the Reviewer's comment, we have now explained in more detail these concepts in the revised manuscript.

4) Some further comments:

1. what would be the scalability of the approach for bigger proteins in terms of number of residues?

There are a few things to consider for molecules of different sizes. The representation requires $3n^2 + 2n + 7e$ floating numbers per input conformation, where n is the number of atoms and e is the number of hyperedges. Each message passing layer of the network requires a constant number of space to store the weights which does not depend on the size of the molecules, but requires $O(e+n)$ time to operate, meaning it scales linearly with respect to size of the molecule. The pooling layer requires $k * (e+n)$ space and $O(k*(e+n)^2)$ time, where k is the number of interest points. Assuming k is much smaller than n and e , each pooling operation scales quadratically with the molecule size.

2. The other aspect to consider is how long it takes for training to converge, which is not theoretically predictable and must be observed empirically for a specific dataset.

Prompted by the Reviewer's comment, we have added a dedicated Section 4.5 entitled "Scalability of the neural network operations" in the Methods.

3. what is really the input needed for this method? Is it a free energy simulation that has visited everywhere? Is it a collection of PDBs? Is it coarse-grained MD? Please clarify cleanly.

The input used to train the neural network comprises a sequence of molecule's conformations obtained from a metadynamics all-atom simulation, each labelled with the free-energy value computed through metadynamics. Each conformation is also endowed with bonded and non-bonded values describing multi-atom interactions used by the hypergraph representation. All this information is stored in a .json file that is the input of the neural network.

4. Finally the english quality is quite poor. In addition technical terms are used very causally - for instance, AUC in abstract.

We have largely revised the manuscript and significantly improved its readability.

We thank you all again for your work and the useful suggestions. We believe that the manuscript is now suitable for publication in *Communications Chemistry*.

Looking forward to hearing from you.

Sincerely,

Vittorio Limongelli

REVIEWERS' COMMENTS:

Reviewer #1 (Remarks to the Author):

The authors have addressed all of my previous concerns satisfactorily.

The methods section is much clearer now and easier to follow. The discussion section also portrays a more complete scenario including the applicability and shortcoming of the current method.

Overall, the revised version is substantially improved and reads smoothly.

I recommend it for publication in its current form.

Reviewer #3 (Remarks to the Author):

I commend the authors for making effort to improve the manuscript. Unfortunately I am still not convinced that even the revised work meets the standards of rigor, usefulness and novelty that would warrant publication.